# Efficient Parallel Training Methods for Spiking Neural Networks with Constant Time Complexity

**Wanjin Feng** [1 2]  **Xingyu Gao\*** [1 2]  **Wenqian Du** [1 2]  **Hailong Shi** [1]  **Peilin Zhao** [3]  **Pengcheng Wu** [4]  **Chunyan Miao** [4]

## Abstract

Spiking Neural Networks (SNNs) often suffer from high time complexity $O(T)$ due to the sequential processing of $T$ spikes, making training computationally expensive. In this paper, we propose a novel Fixed-point Parallel Training (FPT) method to accelerate SNN training without modifying the network architecture or introducing additional assumptions. FPT reduces the time complexity to $O(K)$, where $K$ is a small constant (usually $K = 3$), by using a fixed-point iteration form of Leaky Integrate-and-Fire (LIF) neurons for all $T$ timesteps. We provide a theoretical convergence analysis of FPT and demonstrate that existing parallel spiking neurons can be viewed as special cases of our proposed method. Experimental results show that FPT effectively simulates the dynamics of original LIF neurons, significantly reducing computational time without sacrificing accuracy. This makes FPT a scalable and efficient solution for real-world applications, particularly for long-term tasks. Our code will be released at https://github.com/WanjinVon/FPT.

## 1. Introduction

SNNs represent a biologically inspired evolution of artificial neural networks (ANNs) (Zhu et al., 2024; Zheng et al., 2024). Unlike traditional ANNs that rely on continuous-value propagation, SNNs utilize discrete spikes, mimicking the way the brain processes information (Wang et al., 2024b). This unique spike-based computation offers several advantages, including improved energy efficiency through sparse activation, robustness to noise, and the ability to process

---
[1]Institute of Microelectronics, Chinese Academy of Sciences, Beijing, China [2]University of Chinese Academy of Sciences, Beijing, China [3]School of Artificial Intelligence, Shanghai Jiao Tong University, Shanghai, China [4]Nanyang Technological University, Singapore. Correspondence to: Xingyu Gao <gxy9910@gmail.com>.

*Proceedings of the 42$^{nd}$ International Conference on Machine Learning*, Vancouver, Canada. PMLR 267, 2025. Copyright 2025 by the author(s).

spatiotemporal data effectively (Ma et al., 2023; Singh et al., 2022; Cao et al., 2015). Consequently, SNNs have emerged as a promising bridge between neuroscience and computational science, gaining significant research interest in recent years (Zhang et al., 2023b; Yin et al., 2023).

Despite their potential, SNNs often require a large number of timesteps to achieve optimal performance (Ding et al., 2024). For instance, neuromorphic benchmark datasets such as HAR-DVS, DVS-CIFAR10, and DVS-Gesture typically need 10 or more timesteps to reach satisfactory accuracy (Wang et al., 2024a; Zhuge et al., 2024; Jiang et al., 2024). While longer timesteps enable the network to capture richer temporal information and improve accuracy, they also introduce significant computational overhead (Zhang et al., 2023a). The sequential processing of spikes across $T$ timesteps increases simulation time, slowing down both training and inference, resulting in a time complexity of $O(T)$ (Wang et al., 2023b; Kim et al., 2023b). Moreover, traditional SNN training using BackPropagation Through Time (BPTT) struggles to fully utilize the parallel processing capabilities of modern hardware, such as GPUs, exacerbating computational inefficiencies (Stan & Rhodes, 2024; Hu et al., 2024). This limitation becomes especially pronounced in long-term tasks (Yao et al., 2023).

To address these challenges, we introduce the Fixed-point Parallel Training (FPT) method, which leverages the parallel processing capabilities of modern hardware to significantly accelerate SNN training. By employing a fixed-point iteration framework, FPT decouples sequential dependencies, enabling simultaneous computation across all $T$ timesteps. This reduces the time complexity from $O(T)$ to $O(K)$, where $K$ is the number of fixed-point iterations and typically a small constant. Importantly, FPT preserves essential neural dynamics, including the reset mechanism, ensuring both accuracy and biological interpretability during training. We provide a theoretical analysis to prove the convergence of FPT and demonstrate that existing parallel spiking neuron models can be interpreted as specific instances of our framework. Experimentally, FPT achieves better performance while significantly reducing computational time, making it a scalable and efficient solution for real-world applications.

The main contributions of this paper are as follows:

- We propose a novel Fixed-point Parallel Training (FPT) method that reduces the training time complexity of SNNs from $O(T)$ to $O(K)$, enabling efficient parallel processing while preserving all neural dynamics.

- We provide a theoretical analysis proving the convergence of FPT and demonstrate that existing parallel spiking neuron models can be derived as special cases of our method.

- Experimental results demonstrate that FPT retains the dynamic properties of original LIF neurons, and significantly reduces computational time, addressing the bottlenecks of long-term SNN training.

## 2. Related Work

### 2.1. Training Acceleration for SNNs

Various approaches have been proposed to reduce the computational complexity in SNNs. One common strategy is to reduce the number of timesteps. This can be achieved by dynamically adjusting the number of timesteps based on input samples using confidence score thresholds (Li et al., 2023a;b), dividing SNNs into multiple stages with progressively shrinking timesteps (Ding et al., 2024), or introducing stochastic latency training (Anumasa et al., 2024). Another approach is online training, where the gradients of backpropagation are truncated or approximated along the temporal axis (Xiao et al., 2022; Meng et al., 2022; Jiang et al., 2024). A more direct method involves reducing surrogate gradients from multi-timestep to a single timestep (Suetake et al., 2023). Additionally, SSF stabilizes input and output activations by averaging them over time, reducing gradient computation along the time dimension (Wang et al., 2023a). T-RevSNN accelerates training by deactivating the temporal dynamics of most spiking neurons and introducing multi-level reversible interactions (Hu et al., 2024).

However, these methods often compromise accuracy (e.g., timestep compression, online training), rely on specific network modifications (e.g., T-RevSNN) or assumptions about stabilized spiking flow (e.g., SSF), and therefore lack generality, limiting their scalability to diverse tasks.

### 2.2. Parallel Spiking Neurons

Other approaches focus on developing spiking neuron models that enable parallel computation. For instance, by leveraging the absolute refractory period (ARP) of neurons, the adaptive LIF model achieves constant sequential complexity over the ARP simulation length (Taylor et al., 2023). By separating the linear integration component from the non-linear spiking function, SPSN allows parallel computation across timesteps (Yarga & Wood, 2023). The PSN model further enhances parallelizability by eliminating the

reset mechanism, leading to faster simulation speeds (Fang et al., 2023). Furthermore, PSU and its derivatives, IPSU and RPSU, facilitate parallel computation by decoupling integration and firing mechanisms (Li et al., 2024).

Despite their efficiency, these methods often rely on assumptions, such as the presence of a refractory period or the removal of the reset mechanism, which fail to fully capture the dynamic behavior of LIF neurons. Moreover, models like PSN and PSU introduce additional $O(T^2)$ complexity in learning parameters to maintain accuracy, making it difficult to adapt them for sequential models that handle variable-length sequences.

### 2.3. Fixed-point in Neural Networks

Fixed-point is crucial in neural network models, particularly in implicit layers and recurrent networks, where it occurs when outputs stabilize after multiple iterations (Bai et al., 2019; El Ghaoui et al., 2021). Several studies have integrated fixed-point conditions into RNNs to enhance training efficiency and network stability (Wang & Ragni, 2021; Zhu & Rosenbaum, 2024; Lim et al., 2024). In SNNs, feedback connections cause the average firing rates to evolve toward an equilibrium state. Implicit differentiation of this equilibrium equation allows gradient computation without explicitly tracking the forward process (Xiao et al., 2021; 2023). However, this method requires a sufficient number of timesteps for the model to reach equilibrium, and the model's expressive power is influenced by the depth of the weight-tied block. Recent work, such as Cao et al. (2025), introduces MPIS-SNNs, which also can be viewed as a weight-tied block that leverages fixed-point theory.

Despite these advancements, fixed-point iterative training methods in SNNs still heavily rely on specific network architectures. In particular, the possibility of reformulating the dynamics of LIF neurons into a fixed-point iteration framework—allowing parallel training across timesteps—has not been explored. As a result, achieving an architecture-independent fixed-point parallel training method for SNNs remains an open problem.

## 3. Motivation

Spiking neurons are the fundamental components of SNNs. Among these, Leaky Integrate-and-Fire (LIF) neurons are widely used due to their simplicity in simulating the neuronal behavior. The dynamics of an LIF neuron are described by:

$$u_t = \lambda(u_{t-1} - V_{th}s_{t-1}) + c_t, \qquad (1)$$

where $u_t$ is the membrane potential at timestep $t$, $\lambda < 1$ is the decay factor, $V_{th}$ is the threshold potential, $s_t$ is the binary spike output, and $c_t$ is the synaptic current. When the

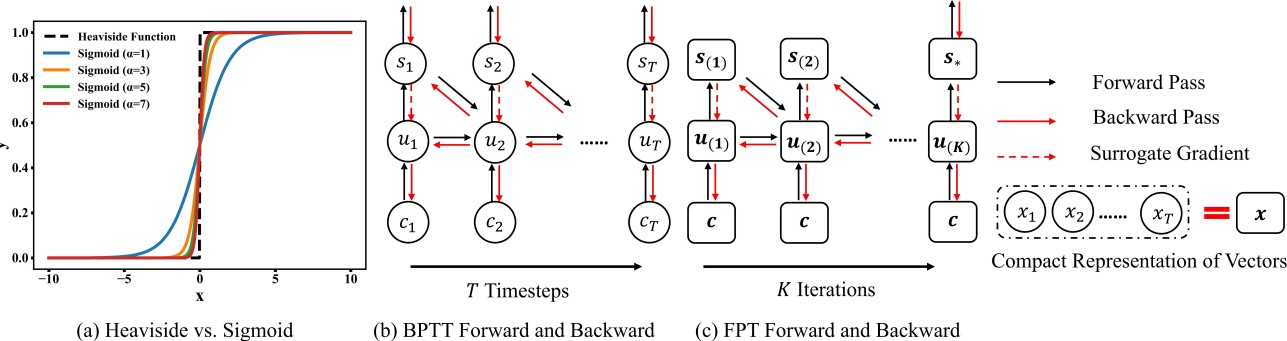

(a) Heaviside vs. Sigmoid   (b) BPTT Forward and Backward   (c) FPT Forward and Backward

*Figure 1.* Comparison of the forward and backward procedures of BPTT and FPT. In BPTT, the neuron processes sequentially over $T$ timesteps, computing step-by-step through time. In contrast, FPT processes all timesteps simultaneously in $K$ iterations, where $K \ll T$. Circles represent scalars, and rounded rectangles represent vectors.

membrane potential exceeds $V_{th}$, the neuron fires a spike:

$$s_t = H(u_t - V_{th}), \qquad (2)$$

where $H(\cdot)$ is the Heaviside step function. LIF neurons process inputs sequentially over discrete timesteps, as shown in Figure 1(b).

To train SNNs, BPTT is used to backpropagate gradients through the inverse process of Eq. (1) and (2). The gradients for $T$ timesteps are computed as:

$$
\begin{aligned}
\frac{\partial \ell}{\partial \mathbf{w}} &= \sum_{t=1}^{T} \frac{\partial \ell}{\partial s_t} \frac{\partial s_t}{\partial u_t} \left( \frac{\partial u_t}{\partial \mathbf{w}} \right. \\
&\left. + \sum_{\tau < t} \prod_{i=\tau}^{t-1} \left( \frac{\partial u_{i+1}}{\partial u_i} + \frac{\partial u_{i+1}}{\partial s_i} \frac{\partial s_i}{\partial u_i} \right) \frac{\partial u_\tau}{\partial \mathbf{w}} \right),
\end{aligned} \qquad (3)
$$

where $\mathbf{w}$ represents the network weights. However, $\frac{\partial s_t}{\partial u_t}$ is non-differentiable due to the discontinuity of $H(\cdot)$. To address this, the surrogate gradient method is applied, using functions like the sigmoid:

$$S_\alpha(u) = \frac{1}{1 + e^{-\alpha u}}, \qquad (4)$$

where $\alpha$ controls the steepness of the approximation. As $\alpha \to \infty$, $S_\alpha(\cdot)$ approaches $H(\cdot)$, as shown in Figure 1(a). The gradient of $H(\cdot)$ can be approximated as:

$$\frac{\partial s}{\partial u} \approx \frac{\partial S_\alpha(u)}{\partial u} = \alpha S_\alpha(u)(1 - S_\alpha(u)), \qquad (5)$$

which smooths the optimization landscape, facilitating effective weight updates while preserving the spiking behavior of the neuron.

Consequently, training and inference in SNNs scale linearly with the number of timesteps $T$, resulting in a time complexity of $O(T)$. This sequential nature poses challenges for real-world applications requiring long temporal dependencies or rapid decision-making.

## 4. FPT: Fixed-point Parallel Training

In this section, we introduce Fixed-point Parallel Training (FPT) method, a novel approach aimed at improving the efficiency of SNN training by leveraging fixed-point iterations and parallel processing.

### 4.1. Fixed-point Mapping

To accelerate the training process of SNNs, a straightforward approach is to unroll the states across all timesteps into vectors or matrices for parallel processing, thereby reducing the time complexity. Specifically, assume the initial membrane potential $u_0 = 0$, the recursive updates for the membrane potential over $T$ timesteps can be approximated as follows:

$$
\begin{cases}
u_1 = c_1 \\
u_2 = \lambda (u_1 - s_1 V_{th}) + c_2 \\
u_3 = \lambda (u_2 - s_2 V_{th}) + c_3 \\
\quad \vdots \\
u_T = \lambda (u_{T-1} - s_{T-1} V_{th}) + c_T
\end{cases} \qquad (6)
$$

Introducing vector and matrix notation, we define:

$$
\mathbf{u} = \begin{pmatrix} u_1 \\ u_2 \\ \vdots \\ u_T \end{pmatrix}, \quad \mathbf{s} = \begin{pmatrix} s_1 \\ s_2 \\ \vdots \\ s_T \end{pmatrix}, \quad \mathbf{c} = \begin{pmatrix} c_1 \\ c_2 \\ \vdots \\ c_T \end{pmatrix}, \qquad (7)
$$

where $\mathbf{u}$ represents the membrane potentials at different timesteps, $\mathbf{s}$ is the vector of spike outputs, and $c$ denotes the synaptic currents over time. To capture the influence of past

inputs, we introduce the decay matrix $\mathbf{\Lambda}$:

$$\mathbf{\Lambda} = \begin{pmatrix} \lambda^0 & 0 & \cdots & 0 \\ \lambda^1 & \lambda^0 & \cdots & 0 \\ \vdots & \vdots & \ddots & \vdots \\ \lambda^{T-1} & \lambda^{T-2} & \cdots & \lambda^0 \end{pmatrix}. \qquad (8)$$

Here, $\mathbf{\Lambda}$ is a lower triangular matrix, where each element in the $i$-th row and $j$-th column is $\lambda^{i-j}$ for $i \geq j$ and 0 otherwise. This matrix models the exponential decay of the past inputs on the current membrane potential, with $\lambda$ being the decay factor.

Using these notations, the membrane potential across all timesteps for parallel iteration of LIF neurons can be compactly represented as:

$$\begin{cases} \mathbf{u} = -V_{th}(\mathbf{\Lambda} - \mathbf{I})\mathbf{s} + \mathbf{\Lambda}\mathbf{c}, \\ \mathbf{s} = H(\mathbf{u} - V_{th}). \end{cases} \qquad (9)$$

The corresponding fixed-point mapping is:

$$\Phi(\mathbf{u}) = -V_{th}(\mathbf{\Lambda} - \mathbf{I})H(\mathbf{u} - V_{th}) + \mathbf{\Lambda}\mathbf{c} \qquad (10)$$

However, this cannot be directly used in optimization algorithms due to the discontinuity introduced by the Heaviside function $H(\cdot)$.

### 4.2. Surrogate Fixed-point Learning

To ensure smooth convergence of the fixed-point iterative equation, we replace the Heaviside function $H(\cdot)$ with a surrogate function, such as the sigmoid approximation:

$$\Phi(\mathbf{u}) \approx \hat{\Phi}_\alpha(\mathbf{u}) = -V_{th}(\mathbf{\Lambda} - \mathbf{I})S_\alpha(\mathbf{u} - V_{th}) + \mathbf{\Lambda}\mathbf{c} \qquad (11)$$

As $\alpha \to \infty$, $\hat{\Phi}_\alpha(\cdot)$ converges to $\Phi(\cdot)$. However, for large values of $\alpha$, the function becomes too steep, which may hinder training due to sharp transitions. Drawing from the concept of surrogate gradients, we use different approximation factors $\alpha$ for the forward and backward passes. Specifically, we use $\alpha_f$ for the forward pass and $\alpha_b$ for the backward pass, ensuring $\alpha_b \leq \alpha_f$. This ensures that the forward pass provides a sufficiently approximation to $\Phi(\cdot)$, while maintaining a stable gradient in the backward pass, facilitating more efficient learning.

### 4.3. Parallel Training

#### 4.3.1. FORWARD PASS

During the forward pass, assuming convergence after $K$ iterations, the membrane potential vector $\mathbf{u}_{(K)}$ converges to an equilibrium point, denoted as $\mathbf{u}_*$, where further iterations produce negligible changes. The iterative forward

---

**Algorithm 1** Forward Pass of FPT

**Require:** Input current $\mathbf{c}$, threshold potential $V_{th}$, decay factor $\lambda$, timesteps $T$
**Ensure:** Membrane potentials $\mathbf{u}_*$, spike outputs $\mathbf{s}_*$
1: Initialize $\mathbf{u}_0 = \mathbf{s}_{(0)} = \mathbf{0}$
2: Compute $\mathbf{\Lambda}$ based on Eq. (8).
3: **for** $k = 1$ to $K$ **do**
4:     Compute $\mathbf{u}_{(k)}$ using:

$$\mathbf{u}_{(k)} = -V_{th}(\mathbf{\Lambda} - \mathbf{I})\mathbf{s}_{(k-1)} + \mathbf{\Lambda}\mathbf{c}$$

5:     Compute $\mathbf{s}_{(k)}$ using:

$$\mathbf{s}_{(k)} = S_{\alpha_f}(\mathbf{u}_{(k)} - V_{th})$$

6: **end for**
7: Set equilibrium membrane potential: $\mathbf{u}_* = \mathbf{u}_{(k)}$
8: Compute spike outputs $\mathbf{s}_*$ based on Eq. (14) or (15)

---

propagation process is described as:

$$\begin{cases} \mathbf{u}_{(1)} = \mathbf{\Lambda}\mathbf{c}, \\ \mathbf{u}_{(2)} = -V_{th}(\mathbf{\Lambda} - \mathbf{I})\mathbf{s}_{(1)} + \mathbf{\Lambda}\mathbf{c}, \\ \quad\vdots \\ \mathbf{u}_{(K)} = -V_{th}(\mathbf{\Lambda} - \mathbf{I})\mathbf{s}_{(K-1)} + \mathbf{\Lambda}\mathbf{c}, \end{cases} \qquad (12)$$

$$\mathbf{s}_{(k)} = S_{\alpha_f}(\mathbf{u}_{(k)} - V_{th}), \quad 1 \leq k \leq K. \qquad (13)$$

Here, $\mathbf{s}_{(k)}$ represents the intermediate spike outputs during the iterative process. As shown in Figure 1(c), this process preserves the LIF neuron dynamics, particularly the reset mechanism, which is absent in models like PSN (Fang et al., 2023).

Upon convergence, the equilibrium membrane potential $\mathbf{u}_*$ is computed, and the final spike outputs $\mathbf{s}_*$ can be determined as follows:

$$\mathbf{s}_* = H(\mathbf{u}_* - V_{th}). \qquad (14)$$

Alternatively, a probabilistic firing mechanism that samples once can also be used, as proposed in Ma et al. (2023):

$$\mathbf{s}_* \sim Bernoulli(S_{\alpha_f}(\mathbf{u}_* - V_{th})). \qquad (15)$$

The forward propagation process is summarized in Algorithm 1. It is worth noting that this forward algorithm can be further improved in several ways, such as early termination of iterations upon convergence or adapting $\alpha_f$ dynamically during the iterative process. Furthermore, $\alpha_f$ can also be learnable. Detailed implementations and discussions are provided in the Appendix.

#### 4.3.2. BACKWARD PASS

Due to the favorable convergence properties of LIF neurons, we set $K = 3$ for most experiments, resulting in a

minimal number of iterations. Unlike traditional deep equilibrium models (Bai et al., 2019; Cao et al., 2025), which rely on iterative methods to compute the inverse Jacobian at equilibrium, our approach directly employs automatic differentiation with surrogate gradients for backpropagation. This is feasible due to the small value of $K$, simplifying the implementation.

Specifically, the loss gradient with respect to network weights $\mathbf{w}$ is:

$$\frac{\partial \ell}{\partial \mathbf{w}} = \frac{\partial \ell}{\partial \mathbf{s}_*} \frac{\partial \mathbf{s}_*}{\partial \mathbf{u}_*} \frac{\partial \mathbf{u}_*}{\partial \mathbf{w}} = \frac{\partial \ell}{\partial \mathbf{s}_*} \frac{\partial \mathbf{s}_*}{\partial \mathbf{u}_*} \frac{\partial \mathbf{u}_{(K)}}{\partial \mathbf{w}}. \quad (16)$$

Here, whether the spike outputs $\mathbf{s}_*$ are computed based on Eq. (14) or (15), their surrogate gradients are:

$$\frac{\partial \mathbf{s}_*}{\partial \mathbf{u}_*} \leftarrow \frac{\partial S_{\alpha_b}(\mathbf{u}_{(K)-V_{th}})}{\partial \mathbf{u}_{(K)}} \quad (17)$$

For $\frac{\partial \mathbf{u}_{(K)}}{\partial \mathbf{w}}$, it can be computed iteratively over $K$ steps as follows:

$$\begin{cases} \frac{\partial \mathbf{u}_{(1)}}{\partial \mathbf{w}} = \mathbf{\Lambda} \frac{\partial \mathbf{c}}{\partial \mathbf{w}}, \\ \frac{\partial \mathbf{u}_{(2)}}{\partial \mathbf{w}} = -V_{th}(\mathbf{\Lambda} - \mathbf{I}) \frac{\partial \mathbf{s}_{(1)}}{\partial \mathbf{u}_{(1)}} \frac{\partial \mathbf{u}_{(1)}}{\partial \mathbf{w}} + \mathbf{\Lambda} \frac{\partial \mathbf{c}}{\partial \mathbf{w}}, \\ \quad \vdots \\ \frac{\partial \mathbf{u}_{(K)}}{\partial \mathbf{w}} = -V_{th}(\mathbf{\Lambda} - \mathbf{I}) \frac{\partial \mathbf{s}_{(K-1)}}{\partial \mathbf{u}_{(K-1)}} \frac{\partial \mathbf{u}_{(K-1)}}{\partial \mathbf{w}} + \mathbf{\Lambda} \frac{\partial \mathbf{c}}{\partial \mathbf{w}}. \end{cases} \quad (18)$$

The surrogate gradient method is applied as follows:

$$\frac{\partial \mathbf{s}_{(k)}}{\partial \mathbf{u}_{(k)}} \leftarrow \frac{\partial S_{\alpha_b}(\mathbf{u}_{(k)} - V_{th})}{\partial \mathbf{u}_{(k)}} \quad (19)$$

The detailed backward propagation process is summarized in Algorithm 2. The time complexity for both the forward and backward passes of the FPT algorithm is $O(K)$, compared to the traditional $O(T)$ ($K \ll T$). This decoupling from $T$ makes FPT particularly efficient for long-term tasks, enabling extended temporal processing at reduced computational costs. Moreover, automatic differentiation eliminates the need to explicitly construct and invert Jacobian matrices, simplifying implementation and making the approach more practical.

## 5. Theoretical Analysis

In this section, we focus on three key research questions: whether convergence can be guaranteed and at what rate, how FPT relates to other parallel spiking neuron models, and what its computational complexity is.

---

**Algorithm 2** Backward Pass of FPT
**Require:** Input current $\mathbf{c}$, threshold potential $V_{th}$, decay factor $\lambda$, iterations $K$
**Ensure:** Compute the gradient $\frac{\partial \mathbf{s}_*}{\partial \mathbf{w}}$
1: Compute the gradient $\frac{\partial \mathbf{s}_*}{\partial \mathbf{u}_*} \leftarrow \frac{\partial S_{\alpha_b}(\mathbf{u}_{(K)-V_{th}})}{\partial \mathbf{u}_{(K)}}$
2: Initialize the gradient $\frac{\partial \mathbf{u}_{(1)}}{\partial \mathbf{w}} \leftarrow \mathbf{\Lambda} \frac{\partial \mathbf{c}}{\partial \mathbf{w}}$
3: **for** $k = 2$ to $K$ **do**
4:    Compute the gradient for the previous potential:

$$\frac{\partial \mathbf{s}_{(k-1)}}{\partial \mathbf{u}_{(k-1)}} \leftarrow \frac{\partial S_{\alpha_b}(\mathbf{u}_{(k-1)} - V_{th})}{\partial \mathbf{u}_{(k-1)}}$$

5:    Update the gradient for the current iteration:

$$\frac{\partial \mathbf{u}_{(k)}}{\partial \mathbf{w}} \leftarrow -V_{th}(\mathbf{\Lambda} - \mathbf{I}) \frac{\partial \mathbf{s}_{(k-1)}}{\partial \mathbf{u}_{(k-1)}} \frac{\partial \mathbf{u}_{(k-1)}}{\partial \mathbf{w}} + \mathbf{\Lambda} \frac{\partial \mathbf{c}}{\partial \mathbf{w}}$$

6: **end for**
7: Compute the final gradient for the network weights:

$$\frac{\partial \mathbf{s}_*}{\partial \mathbf{w}} \leftarrow \frac{\partial \mathbf{s}_*}{\partial \mathbf{u}_*} \frac{\partial \mathbf{u}_{(K)}}{\partial \mathbf{w}}$$

---

### 5.1. Analysis of Convergence Condition and Speed

**Lemma 5.1.** *Assume the surrogate function $S_\alpha(\cdot)$ is Lipschitz continuous with a constant $L_\alpha$. If the condition $V_{th}L_\alpha \frac{\lambda(1-\lambda^{T-1})}{1-\lambda} < 1$, where $0 < \lambda < 1$, is satisfied, then the mapping $\hat{\Phi}_\alpha(\mathbf{u}) = -V_{th}(\mathbf{\Lambda} - \mathbf{I})S_\alpha(\mathbf{u} - V_{th}) + \mathbf{\Lambda}\mathbf{c}$ is a contraction mapping under the 1-norm. Consequently, the iterative scheme*

$$\mathbf{u}_{(k)} = -V_{th}(\mathbf{\Lambda} - \mathbf{I})S_\alpha(\mathbf{u}_{(k-1)} - V_{th}) + \mathbf{\Lambda}\mathbf{c} \quad (20)$$

*converges to a unique fixed point $\mathbf{u}_*$.*

To illustrate the convergence of the algorithm, consider the typical values $V_{th} = 1$, $\lambda = 0.25$, and the sigmoid function $S_\alpha(\cdot)$, which is Lipschitz continuous with constant $L_\alpha = \frac{\alpha}{4}$. Substituting these values into the convergence condition:

$$V_{th}L_\alpha \frac{\lambda(1 - \lambda^{T-1})}{1-\lambda} = \frac{\alpha(1 - 0.25^{T-1})}{12} \leq \frac{\alpha}{12}, \quad (21)$$

We see that for any $\alpha < 12$, this expression is always less than 1, thereby satisfying the convergence condition. Moreover, the convergence proof considers the worst-case scenario. In practice, the actual convergence condition is more relaxed due to the inherent sparsity of neural activity. Additionally, this convergence condition also reflects the rate of convergence: smaller $\alpha$ leads to faster convergence.

Furthermore, the matrix $\mathbf{\Lambda} - \mathbf{I}$ is strictly lower triangular, with a spectral radius of zero since all its eigenvalues are

*Table 1.* Complexity Comparison of SNN Training Methods

| Training Methods | Memory | Training Time | Inference Energy | Applicability |
|---|---|---|---|---|
| OTTT (Xiao et al., 2022) | $\mathcal{O}(L)$ | $\mathcal{O}(LT)$ | $\mathcal{O}(T)$ | Limited |
| SLTT-$k$ (Meng et al., 2023) | $\mathcal{O}(L)$ | $\mathcal{O}(Lk)$ | $\mathcal{O}(T)$ | Limited |
| T-RevSNN turn-off (Hu et al., 2024) | $\mathcal{O}(L)$ | $\mathcal{O}(L)$ | $\mathcal{O}(1)$ | Limited |
| T-RevSNN turn-on (Hu et al., 2024) | $\mathcal{O}(L)$ | $\mathcal{O}(T)$ | $\mathcal{O}(1)$ | Limited |
| BPTT (Zheng et al., 2021) | $\mathcal{O}(LT)$ | $\mathcal{O}(LT)$ | $\mathcal{O}(T)$ | Unlimited |
| **FPT (Ours)** | $\mathcal{O}(LT) + \lambda\mathcal{O}(LKT)$ | $\mathcal{O}(LK)$ | $\mathcal{O}(T)$ | Unlimited |

zero. This property ensures rapid convergence, as the iterative process does not amplify errors through eigenvalues, thereby maintaining stability and guaranteeing efficient convergence. Importantly, Eq. (21) shows that the convergence rate (or condition) can be decoupled from the timesteps $T$ by approximating the term $0.25^{T-1}$ as 0. This indicates that the iteration is weakly influenced by $T$, ensuring rapid convergence even for longer timesteps.

### 5.2. Comparison with Parallel Spiking Neuron

By modifying FPT slightly, we can adapt it to learnable LIF neurons, with the update rule in Eq. (9) incorporating learnable parameters as follows:

$$\begin{cases} \mathbf{u} = -V_{th}(\mathbf{A} - \mathbf{I})\mathbf{s} + \mathbf{Ac} \\ \mathbf{s} = H(\mathbf{u} - \mathbf{B}) \end{cases} \tag{22}$$

Here, $\mathbf{A} \in \mathbb{R}^{T \times T}$ is the learnable decay matrix, and $\mathbf{B} \in \mathbb{R}^T$ is the learnable threshold vector. This model can be trained using a forward and backward propagation algorithm similar to those in Eq. (12) and (18).

For a single fixed-point iteration, the update for the membrane potential and spike output can be expressed as:

$$\begin{cases} \mathbf{u} = \mathbf{Ac} \\ \mathbf{s} = H(\mathbf{u} - \mathbf{B}). \end{cases} \tag{23}$$

This formulation represents the standard Parallel Spiking Neuron (PSN) (Fang et al., 2023). Unlike the traditional LIF model, PSN removes the reset mechanism, which can affect the algorithm's accuracy and biological fidelity.

By setting the threshold vector in Eq. (22) to a fixed value $V_{th}$ and performing two iterations, the Parallel Spiking Unit (PSU) model can be derived (Li et al., 2024):

$$\begin{cases} \mathbf{u} = -V_{th}(\mathbf{A} - \mathbf{I})S_\alpha(\mathbf{Ac} - V_{th}) + \mathbf{Ac} \\ \mathbf{s} = H(\mathbf{u} - V_{th}), \end{cases} \tag{24}$$

which further simplifies to:

$$\mathbf{s} = H(\mathbf{Ac} - \mathbf{D}S_\alpha(\mathbf{Ac} - V_{th}) - V_{th}) \tag{25}$$

where $\mathbf{D} = V_{th}(\mathbf{A} - \mathbf{I})$. This formulation incorporates feedback mechanisms where the previous state influences the current potential. The term $S_\alpha(\mathbf{Ac} - V_{th})$ represents an estimate of the spike output, used to adjust the membrane potential. Therefore, PSN and PSU can be viewed as intermediate forms of FPT for learnable LIF neurons.

However, it is important to note that convergence is not guaranteed with just one or two iterations. In these cases, the fixed-point iteration may fail to reach an equilibrium point, potentially leading to inaccurate simulations of LIF dynamics and degraded performance. Additionally, PSN and PSU introduce an extra $O(T^2)$ learnable parameter complexity, which limits their applicability to long sequences.

### 5.3. Analysis of Computational Complexity

Table 1 summarizes the theoretical complexity of various SNN training approaches. Let $L$ denote the number of network layers, $k$ the truncated temporal length in SLTT-$k$, and $K$ the number of fixed-point iterations used in FPT. Typically, $K$ is a small constant (e.g., $K = 3$) and remains independent of $T$, making its contribution negligible for large $T$. Thus, space complexity $\mathcal{O}(LKT)$ and time complexity $\mathcal{O}(LK)$ can be approximated as $\mathcal{O}(LT)$ and $\mathcal{O}(L)$, respectively. The coefficient $\lambda$ represents the fraction of memory introduced by the LIF component, which is the only part affected by FPT, while all other components of the network remain unchanged.

Notably, methods like OTTT, SLTT, and T-RevSNN accelerate training by truncating gradients or discarding temporal dependencies. While effective, such strategies limit their ability to model fine-grained temporal dynamics. In contrast, FPT preserves complete neuronal dynamics and enables efficient training without altering the model architecture, thereby offering broader applicability across diverse SNN models.

## 6. Experiments

This section answers three key questions: First, does the equilibrium point of FPT converge to the original LIF dynamics? Second, can FPT improve training speed without

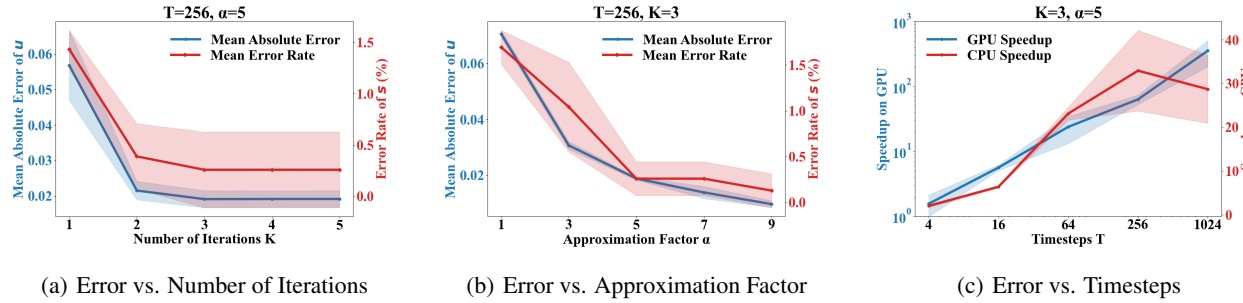

| (a) Error vs. Number of Iterations | (b) Error vs. Approximation Factor | (c) Error vs. Timesteps |

*Figure 2.* Convergence and efficiency of the LIF neuron using our proposed FPT method. The absolute error in membrane potential, spike firing error rate, and speedup factor relative to the original LIF are reported. Solid lines show the mean values across three experiments, with shaded areas representing the standard deviation.

compromising accuracy? Finally, is it necessary to preserve the complete neural dynamics of LIF neurons?

### 6.1. LIF Dynamics Simulation

In this section, we validate whether the equilibrium point of the forward pass in FPT converges to the original LIF dynamics. We used a LIF neuron with a decay coefficient $\lambda = 0.25$, a threshold $V_{th} = 1$, and input currents drawn from a Gaussian distribution with a mean of 0 and a variance of 1. The experiments were conducted on an RTX 3060 laptop GPU and a 12th Gen Intel i7-12700H CPU.

As shown in Figure 2(a), with increasing iterations, the equilibrium point of FPT quickly converges to the original LIF dynamics, including both membrane potential and spike behavior. This demonstrates the high biological fidelity of FPT, retaining the key characteristics of the LIF neuron. For $K \geq 3$, changes in membrane potential and spike activity become negligible, indicating equilibrium. Based on this observation, we chose $K = 3$ for all subsequent experiments. Figure 2(b) highlights the impact of the sigmoid approximation parameter $\alpha$ on convergence. With a fixed number of iterations, increasing $\alpha$ reduces the discrepancy between the parallel and original LIF dynamics, approaching zero. This aligns with Figure 1(a), showing that larger $\alpha$ values improve approximation accuracy. Finally, Figure 2(c) demonstrates the time efficiency of FPT on GPU and CPU platforms. As $T$ increases, the GPU-based FPT achieves linear speedup, outperforming traditional methods. For $T = 512$ or longer, FPT-based LIF models achieve over $100\times$ speedup on GPUs, underscoring the effectiveness of FPT for long-term simulations.

### 6.2. General Classification Performance of FPT

To evaluate the effectiveness of FPT compared to different training methods, we tested it on three datasets: the dynamic DVS-CIFAR10 and DVS-Gesture datasets, and

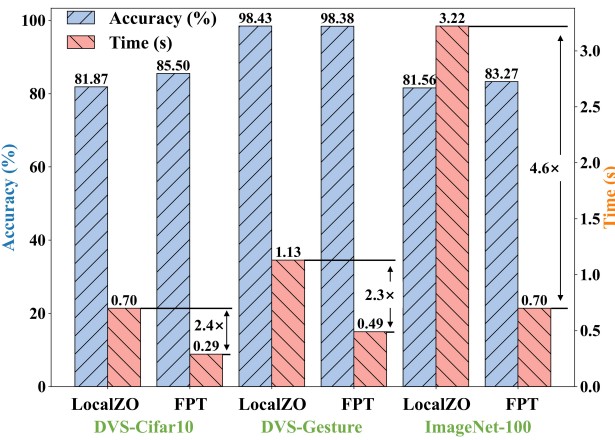

*Figure 3.* Comparison of accuracy and speed between FPT and the BPTT-like algorithm (LocalZO). Time indicates the average training time per batch on a single RTX 3090 GPU, with both models using the same architecture and hyperparameters.

the static ImageNet-100 dataset, as shown in Table 2. Detailed experimental settings are provided in the Appendix. On the DVS-CIFAR10 dataset, FPT achieved an improvement of over $3\%$ in accuracy compared to the baselines. In contrast, methods such as online training and timestep shrinkage, which rely on gradient truncation or reducing timesteps, yielded inferior accuracy. On the DVS-Gesture dataset, our method achieved an accuracy comparable to LocalZO. For the static ImageNet-100 dataset, FPT outperformed LocalZO. This improvement in performance is attributed to the enhanced generalization capability provided by our surrogate fixed-point learning framework.

Figure 3 compares FPT with LocalZO in terms of both accuracy and speed across the datasets. Since FPT reduces time complexity from $O(T)$ to $O(K)$, with $K = 3$, and it also avoids the multiple sampling required by LocalZO. As

*Table 2.* Comparison of FPT with previous SOTA training method. The average accuracy and standard deviation across three runs are reported, with the highest accuracy from the three runs shown in parentheses.

| Dataset | Model | Method | Architecture | Training Timestep | Accuracy (%) |
|---|---|---|---|---|---|
| | OTTT (Xiao et al., 2022) | Online training | VGG-11 | 10 | 76.63 |
| | NDOT (Jiang et al., 2024) | Online training | VGG-11 | 10 | 77.40 |
| | SLTT (Meng et al., 2023) | Timestep Shrinkage | VGG-11 | 10 | 77.17 |
| | SEENN (Li et al., 2023b) | Timestep Shrinkage | VGG-16 | 10 | 82.70 |
| DVS-CIFAR10 | SSNN (Ding et al., 2024) | Timestep Shrinkage | VGG-9 | 8 | 78.57 |
| | SLT (Anumasa et al., 2024) | Timestep Shrinkage | VGG-11 | 10 | 81.46 |
| | SSF (Wang et al., 2023a) | Stabilized Spiking Flow | VGG-11 | 20 | 78.00 |
| | T-RevSNN (Hu et al., 2024) | Temporal Reversible | ResNet-18 | 16 | 79.20 |
| | LocalZO (Mukhoty et al., 2023) | Zeroth Order | VGG-11 | 10 | 81.87 |
| | **FPT (Ours)** | Parallel Training | VGG-11 | $10\ (K=3)$ | $85.50\pm0.22$ (**85.70**) |
| | SLTT (Meng et al., 2023) | Timestep Shrinkage | VGG-11 | 20 | 97.92 |
| | SSNN (Ding et al., 2024) | Timestep Shrinkage | VGG-9 | 8 | 94.91 |
| DVS-Gesture | T-RevSNN (Hu et al., 2024) | Temporal Reversible | ResNet-18 | 16 | 97.90 |
| | LocalZO (Mukhoty et al., 2023) | Zeroth Order | VGG-11 | 10 | 98.43 |
| | **FPT (Ours)** | Parallel Training | VGG-11 | $20\ (K=3)$ | $98.38\pm0.17$ (**98.61**) |
| | EfficientLIF-Net (Kim et al., 2023a) | Normal BPPT | ResNet-19 | 5 | 79.44 |
| ImageNet-100 | LocalZO (Mukhoty et al., 2023) | Zeroth Order | SEW-Resnet34 | 4 | 81.56 |
| | **FPT (Ours)** | Parallel Training | SEW-Resnet34 | $4\ (K=3)$ | $83.27\pm0.16$ (**83.48**) |

a result, our algorithm achieved more than twice the speed of LocalZO while maintaining comparable or even superior performance.

## 6.3. Efficiency Comparison with BPTT

*Table 3.* Comparison of FPT and BPTT with different timesteps $T$. The p-values from t-tests show that accuracy differences are not statistically significant. FPT accuracy is reported across three random trials. Time indicates the average training time per batch on a single RTX 3090 GPU.

| Method | Timesteps | Firing rate (%) | Accuracy (%) | Time (ms) | P-value |
|---|---|---|---|---|---|
| | 8 | $12.24\pm0.01$ | $90.95\pm0.17$ | 6.49 | - |
| BPTT | 32 | $12.29\pm0.07$ | $92.15\pm0.33$ | 21.96 | - |
| | 128 | $12.51\pm0.10$ | $92.66\pm0.34$ | 74.04 | - |
| | 8 | $12.33\pm0.09$ | $90.77\pm0.38$ | 1.55 | 0.512 |
| **FPT** | 32 | $12.41\pm0.13$ | $92.25\pm0.15$ | 2.01 | 0.668 |
| | 128 | $12.61\pm0.04$ | $92.55\pm0.30$ | 5.58 | 0.696 |

To further evaluate the efficiency of FPT across different timesteps $T$, we conducted experiments comparing it with the traditional BPTT on the Amazon Photos dataset using DRSGNN, which incorporates a layer of LIF neurons. The experiments were performed on a single RTX 3090 GPU, with a batch size of 64. Except for the timesteps, all experimental settings followed those in (Zhao et al., 2024).

As shown in Table 3, both FPT and BPTT demonstrate improved accuracy as $T$ increases, indicating that longer timesteps allow the model to capture more temporal dynamics and achieve better performance. However, the t-tests reveal that the differences in accuracy between FPT and BPTT are not statistically significant, with p-values consis-

tently greater than 0.05. Furthermore, regardless of whether BPTT or FPT is used, the firing rate remains around 12% across different timesteps. Notably, FPT exhibits a significant efficiency advantage. For $T \geq 32$, FPT is over 10 times faster than BPTT. This demonstrates FPT's ability to dramatically accelerate training while maintaining performance comparable to that of BPTT.

## 6.4. Ablation Study

As introduced in Section 5.2, PSN and IPSU can be viewed as intermediate results of our FPT with different values of $K$. At the same time, they diminish the impact of the reset mechanism, with PSN removing it entirely. To evaluate the impact of the number of iterations $K$ and the reset mechanism, we conducted an ablation study comparing various configurations of FPT, PSN, and IPSU on the sequential CIFAR10 and CIFAR100 datasets. In these tasks, images are processed column by column over 32 timesteps, and the experimental settings follow those in (Fang et al., 2023).

As shown in Table 4, when the decay matrix $\mathbf{A}$ is fixed and non-learnable, $\text{PSN}_0$ exhibits approximately 2% lower accuracy compared to LIF across both datasets, which aligns with previous results reported for PSN (Fang et al., 2023). The key distinction here is that PSN removes the reset mechanism, which is an essential feature in the traditional LIF model. Akin to the forget gate in LSTM networks, it regulates the retention of historical information by discarding past states after each spike.

In contrast, when the decay matrix $\mathbf{A}$ and the firing threshold $\mathbf{B}$ are learnable, FPT consistently outperforms both PSN

*Table 4.* Comparison on the sequential CIFAR datasets. The subscript $_0$ and symbol $\times$ indicate that the matrix $\mathbf{A}$ is set to $\mathbf{\Lambda}$ and is non-learnable. $\diagdown\!\!\!\times$ denotes that only the lower triangular part of the matrix $\mathbf{A}$ is learnable to avoid using future information. The table reports the accuracy of FPT across three random trials. Other results are taken from (Fang et al., 2023) and (Li et al., 2024).

| Dataset | Method | Learnable | Accuracy (%) |
|---|---|---|---|
| Sequential CIFAR10 | LIF | $\times$ | 81.50 |
| | $PSN_0$ | $\times$ | 79.80 |
| | **FPT** | $\times$ | 81.54±0.34 |
| | IPSU | $\diagdown\!\!\!\times$ | 87.28 |
| | Masked PSN | $\diagdown\!\!\!\times$ | 85.81 |
| | **FPT w/ Masked A** | $\diagdown\!\!\!\times$ | 87.48±0.14 |
| | PSN | $\checkmark$ | 88.45 |
| | **FPT w/ A** | $\checkmark$ | **89.53±0.13** |
| Sequential CIFAR100 | LIF | $\times$ | 55.45 |
| | $PSN_0$ | $\times$ | 53.12 |
| | **FPT** | $\times$ | 55.19±0.37 |
| | IPSU | $\diagdown\!\!\!\times$ | 59.76 |
| | Masked PSN | $\diagdown\!\!\!\times$ | 60.69 |
| | **FPT w/ Masked A** | $\diagdown\!\!\!\times$ | 62.24 ±0.52 |
| | PSN | $\checkmark$ | 62.21 |
| | **FPT w/ A** | $\checkmark$ | **64.50±0.34** |

and IPSU, achieving optimal results across both datasets. This suggests that FPT with $K = 3$ iterations outperform the PSN and IPSU models, which can be considered as having $K = 1$ or $K = 2$. Additionally, for FPT without learnable parameters, the accuracies on both datasets are nearly identical to those of the original LIF, indicating that $K = 3$ is sufficient to ensure convergence. These findings highlight the advantages of our approach, underscoring the importance of both the reset mechanism and sufficient iterations for maintaining model accuracy.

## 7. Conclusion

In this paper, we introduce the Fixed-point Parallel Training (FPT) method, which significantly improves the efficiency of training SNNs. By leveraging parallel fixed-point iterations of LIF neurons, FPT reduces the time complexity from $O(T)$ to $O(K)$, where $K$ is a small constant, leading to substantial speedup. We demonstrate that FPT preserves the biological dynamics of LIF neurons and effectively converges to them in practical applications. This equivalence allows for seamless conversion between sequential and parallel training modes, making FPT ideal for parallel training and subsequent deployment on neuromorphic devices, where inputs can be processed sequentially. Furthermore, FPT outperforms BPTT in computational efficiency, achieving up to $100\times$ speedup on GPUs for long-term tasks. Importantly, FPT requires no modifications to the network architecture, ensuring its broad applicability to a wide range of SNN models. This makes FPT a highly scalable and adaptable solution for efficient SNN training, especially in applica-

tions where performance, speed, and biological fidelity are critical.

## Acknowledgements

This work was supported in part by Science and Technology Innovation (STI) 2030—Major Projects under Grant 2022ZD0208700, and National Natural Science Foundation of China under Grant 62376264.

## Impact Statement

This paper presents work whose goal is to advance the field of Machine Learning. There are many potential societal consequences of our work, none which we feel must be specifically highlighted here.

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

## A. Convergence Proof for FPT

**Lemma A.1.** *Assume the substitution function $S_\alpha(\cdot)$ is Lipschitz continuous with a constant $L_\alpha$. If the condition $V_{th}L_\alpha \frac{\lambda(1-\lambda^{T-1})}{1-\lambda} < 1$, where $0 < \lambda < 1$, is satisfied, then the mapping $\hat{\Phi}_\alpha(\mathbf{u}) = -V_{th}(\mathbf{\Lambda} - \mathbf{I})S_\alpha(\mathbf{u} - V_{th}) + \mathbf{\Lambda}\mathbf{c}$ is a contraction mapping under the 1-norm. Consequently, the iterative scheme*

$$\mathbf{u}_{(k)} = -V_{th}(\mathbf{\Lambda} - \mathbf{I})S_\alpha(\mathbf{u}_{(k-1)} - V_{th}) + \mathbf{\Lambda}\mathbf{c} \tag{26}$$

*converges to a unique fixed point $\mathbf{u}_*$.*

*Proof.* Let $\mathbf{u}_1, \mathbf{u}_2 \in \mathbb{R}^T$. Considering the 1-norm, we have:

$$\|\hat{\Phi}_\alpha(\mathbf{u}_1) - \hat{\Phi}_\alpha(\mathbf{u}_2)\|_1$$
$$= \| - V_{th}(\mathbf{\Lambda} - \mathbf{I})S_\alpha(\mathbf{u}_1 - V_{th}) + \mathbf{\Lambda}\mathbf{c} - (-V_{th}(\mathbf{\Lambda} - \mathbf{I})S_\alpha(\mathbf{u}_2 - V_{th}) + \mathbf{\Lambda}\mathbf{c})\|_1$$
$$= \| - V_{th}(\mathbf{\Lambda} - \mathbf{I})(S_\alpha(\mathbf{u}_1 - V_{th}) - S_\alpha(\mathbf{u}_2 - V_{th}))\|_1.$$

Since $S_\alpha(\mathbf{u})$ is Lipschitz continuous with a constant $L_H$, we have:

$$\|S_\alpha(\mathbf{u}_1 - V_{th}) - S_\alpha(\mathbf{u}_2 - V_{th})\|_1 \le L_\alpha\|\mathbf{u}_1 - \mathbf{u}_2\|_1. \tag{27}$$

Using Hölder's inequality (Cheung, 2001), we get:

$$\|\hat{\Phi}_\alpha(\mathbf{u}_1) - \hat{\Phi}_\alpha(\mathbf{u}_2)\|_1 \le V_{th}\|(\mathbf{\Lambda} - \mathbf{I})\|_\infty \cdot L_\alpha\|\mathbf{u}_1 - \mathbf{u}_2\|_1 = L\|\mathbf{u}_1 - \mathbf{u}_2\|_1, \tag{28}$$

where $L = V_{th}L_\alpha\|(\mathbf{\Lambda} - \mathbf{I})\|_\infty$.

We use the maximum row sum norm (infinity norm) for $\mathbf{\Lambda} - \mathbf{I}$:

$$\|\mathbf{\Lambda} - \mathbf{I}\|_\infty = \sum_{i=1}^{T-1} \lambda^i = \frac{\lambda(1 - \lambda^{T-1})}{1 - \lambda}. \tag{29}$$

According to the Banach fixed-point theorem, if the condition $L = V_{th}L_H \frac{\lambda(1-\lambda^{T-1})}{1-\lambda} < 1$ is satisfied, then $\hat{\Phi}_\alpha(\mathbf{u})$ is a contraction mapping (Shukla et al., 2016). Consequently, the iterative scheme

$$\mathbf{u}_{(k)} = -V_{th}(\mathbf{\Lambda} - \mathbf{I})S_\alpha(\mathbf{u}_{(k-1)} - V_{th}) + \mathbf{\Lambda}\mathbf{c} \tag{30}$$

converges to a unique fixed point $\mathbf{u}_*$.

$\square$

## B. FPT Variants

In some cases, the LIF neuron does not require $K$ iterations according to Eq. (30) to converge. Therefore, we introduce an early stopping criterion based on the change in membrane potential, which can accelerate the training process, as shown in Algorithm 3. Specifically, if the change in membrane potential between iterations falls below a certain threshold $\epsilon$, the iteration can be terminated early.

Additionally, based on the convergence theorem of FPT, we observe that smaller values of $\alpha_f$ lead to faster convergence, but with lower approximation accuracy. Conversely, larger values of $\alpha_f$ provide more accurate approximations at the cost of slower convergence. A natural strategy is to start the iteration with a smaller $\alpha_f$ for a good initial approximation, and then gradually increase $\alpha_f$ for more accurate approximations, as shown in Algorithm 4, where $\{\alpha_{f1} \le \alpha_{f2} \le \cdots \le \alpha_{fK}\}$. Alternatively, $\alpha_f$ could also be treated as a learnable parameter during training to dynamically adjust the balance between convergence speed and approximation accuracy.

---

**Algorithm 3** Forward Pass of FPT with Early Stop for **u**

---

**Require:** Input current $\mathbf{c}$, threshold potential $V_{th}$, decay factor $\lambda$, timesteps $T$, tolerance $\epsilon$
**Ensure:** Membrane potentials $\mathbf{u}_*$, spike outputs $\mathbf{s}_*$
 1: Initialize $\mathbf{u}_0 = \mathbf{s}_{(0)} = \mathbf{0}$
 2: Compute $\mathbf{\Lambda}$ based on Eq. (8).
 3: Set $k = 1$
 4: **while** $k \leq K$ **do**
 5:     Compute $\mathbf{u}_{(k)}$ using:
$$\mathbf{u}_{(k)} = -V_{th}(\mathbf{\Lambda} - \mathbf{I})\mathbf{s}_{(k-1)} + \mathbf{\Lambda}\mathbf{c}$$
 6:     Compute $\mathbf{s}_{(k)}$ using:
$$\mathbf{s}_{(k)} = S_{\alpha_f}(\mathbf{u}_{(k)} - V_{th})$$
 7:     **if** $\|\mathbf{u}_{(k)} - \mathbf{u}_{(k-1)}\|_2 < \epsilon$ **then**
 8:         Break (early stop)
 9:     **end if**
10:     Increment $k \leftarrow k + 1$
11: **end while**
12: Set equilibrium membrane potential: $\mathbf{u}_* = \mathbf{u}_{(k)}$
13: Compute spike outputs $\mathbf{s}_*$ based on Eq. (14) or (15)

---

---

**Algorithm 4** Forward Pass of FPT with Adaptive $\alpha_f$

---

**Require:** Input current $\mathbf{c}$, threshold potential $V_{th}$, decay factor $\lambda$, timesteps $T$, list of $\alpha_f$ values $\{\alpha_{f1}, \alpha_{f2}, \ldots, \alpha_{fK}\}$
**Ensure:** Membrane potentials $\mathbf{u}_*$, spike outputs $\mathbf{s}_*$
 1: Initialize $\mathbf{u}_0 = \mathbf{s}_{(0)} = \mathbf{0}$
 2: Compute $\mathbf{\Lambda}$ based on Eq. (8).
 3: Set $k = 1$
 4: **while** $k \leq K$ **do**
 5:     Set $\alpha_f = \alpha_{fk}$    (select $\alpha_f$ from the predefined list)
 6:     Compute $\mathbf{u}_{(k)}$ using:
$$\mathbf{u}_{(k)} = -V_{th}(\mathbf{\Lambda} - \mathbf{I})\mathbf{s}_{(k-1)} + \mathbf{\Lambda}\mathbf{c}$$
 7:     Compute $\mathbf{s}_{(k)}$ using:
$$\mathbf{s}_{(k)} = S_{\alpha_f}(\mathbf{u}_{(k)} - V_{th})$$
 8:     Increment $k \leftarrow k + 1$
 9: **end while**
10: Set equilibrium membrane potential: $\mathbf{u}_* = \mathbf{u}_{(k)}$
11: Compute spike outputs $\mathbf{s}_*$ based on Eq. (14) or (15)

---

## C. Experimental Setup

For the FPT experiment on the Amazon Photos dataset using DRSGNN, we set $K = 3$ and used Eq. (14) as the firing function.

Table 5 provides the hyperparameter settings corresponding to Table 1 in the main text. The parameter $\alpha_f$ represents the incrementally increasing parameter used during each iteration. According to Lemma 5.1, when $\alpha_f$ is relatively small, it facilitates convergence during the first iteration. As the number of iterations increases, $\alpha_f$ gradually approaches the step function used by the LIF neuron. In the backward pass, $\alpha_b = \alpha_f/3$, which is one-third of the forward pass value, helps to obtain a smoother gradient and avoid vanishing gradients.

As described in the PSN paper, the CIFAR dataset's images are processed by the SNN one column at a time, similar to how humans read from left to right. The corresponding hyperparameters are listed in Table 6.

*Table 5.* Training parameters for FPT on various datasets. Optimizer: Adam with betas: (0.9, 0.999), Rate Scheduler: cosine annealing.

|  | DVS-CIFAR-10 | DVS-Gesture | ImageNet-100 |
|---|---|---|---|
| Number epochs | 300 | 200 | 300 |
| Mini batch size | 32 | 32 | 64 |
| T | 10 | 20 | 4 |
| $\lambda$ | 0.5 | 0.5 | 0.5 |
| $u_0$ | 0 | 0 | 0 |
| $V_{th}$ | 1 | 1 | 1 |
| $\alpha_f$ | 3, 12, 12 | 3, 12, 12 | 2, 6, 12 |
| $\alpha_b$ | $\alpha_f/3$ | $\alpha_f/3$ | $\alpha_f/3$ |
| K | 3 | 3 | 3 |
| $\lambda_{\text{TET}}$ | 0.05 | 0.05 | 1 |
| Learning Rate | 0.1 | 0.1 | 0.005 |
| Optimizer | SGD | SGD | SGD |
| Firing Mechanism | Probabilistic firing | Probabilistic firing | Probabilistic firing |

*Table 6.* Training parameters for FPT on Sequential CIFAR datasets.

|  | Sequential CIFAR10 | Sequential CIFAR100 |
|---|---|---|
| Number epochs | 256 | 256 |
| Mini batch size | 128 | 128 |
| T | 32 | 32 |
| $\lambda$ | 0.5 | 0.5 |
| $u_0$ | 0 | 0 |
| $u_{th}$ | 1 | 1 |
| $\alpha_f$ | 1, 3, 12 | 1, 3, 12 |
| $\alpha_b$ | $\alpha_f/3$ | $\alpha_f/3$ |
| K | 3 | 3 |
| Learning Rate | 0.1 | 0.1 |
| Optimizer | SGD | SGD |
| Firing Mechanism | Probabilistic firing | Probabilistic firing |

*Table 7.* Comparison between BPTT and FPT at Different Simulation Timesteps $T$

| Method | $T$ | Training Time (s) | Inference Time (s) | Memory (MB) | Accuracy (%) |
|---|---|---|---|---|---|
| BPTT | 8 | 0.0195 | 0.0042 | 600 | $97.75 \pm 0.16$ |
|  | 64 | 0.0835 | 0.0257 | 956 | $97.67 \pm 0.17$ |
|  | 512 | 1.701 | 0.2092 | 3238 | $97.84 \pm 0.12$ |
| **FPT** | 8 | 0.0096 | 0.0021 | 622 | $97.73 \pm 0.23$ |
|  | 64 | 0.0109 | 0.0021 | 1162 | $97.71 \pm 0.11$ |
|  | 512 | 0.0803 | 0.0021 | 4264 | $97.70 \pm 0.22$ |

## D. Training and Inference Complexity Comparison

Our experiments focus on comparing the complexity of BPTT and FPT at different timesteps $T$, as shown in Table 7. We conducted experiments on the MNIST dataset using a 3-layer MLP (784×256×128×10), with a batch size of 256, a learning rate of 0.001, and 80 epochs.

Here, "Time" refers to the average running time per batch during training or inference on a single A100 GPU. As shown,

both training and inference times for BPTT increase approximately linearly with $T$. In contrast, FPT's training time increases only slightly, while its inference time remains nearly constant. At $T = 256$, FPT achieves a 21× speedup over BPTT during training.

FPT introduces only a slight increase in memory usage due to the LIF activation function during training, without impacting other network components. For larger networks such as MS-ResNet-18, the relative increase in memory consumption is even smaller.

## E. Network Dynamics Across Long Timesteps

Table 8. Cosine Similarity (%) between Original and FPT Outputs under Different $\alpha$ and Simulation Timesteps $T$

| $\alpha$ | $T = 8$ | $T = 64$ | $T = 512$ |
|---|---|---|---|
| 5 | 99.89 | 99.55 | 99.53 |
| 7 | 99.90 | 99.49 | 99.47 |

We replaced the LIF neurons in a pre-trained 3-layer LIF-based MLP (784×256×128×10) trained on the MNIST dataset with parallel LIF neurons based on FPT. Table 8 shows the cosine similarity (%) between the original and FPT-replaced outputs for $T$=8, 64, and 512.

As $T$ increases, the outputs of the FPT-based parallel LIF and the original LIF neurons show slight divergence due to error accumulation, reducing the similarity of the final network outputs. However, even at $T = 512$, the similarity remains above 99.5%, demonstrating that FPT preserves a high degree of consistency in network dynamics. Furthermore, this minor discrepancy can be effectively mitigated through light fine-tuning.

## F. Learnable Decay Matrix Evaluation

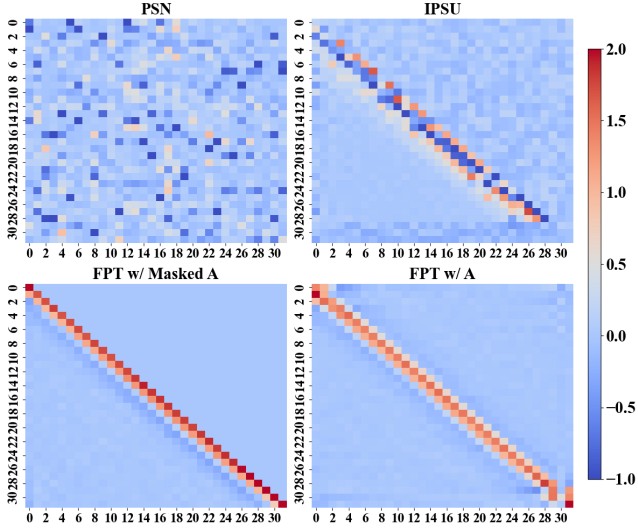

Figure 4. Learned decay matrices for different parallel spiking neurons.

Figure 4 illustrates the final learned decay matrices of different parallel spiking neurons on the Sequential CIFAR10 dataset. Although PSN achieved an accuracy of 88.45% on this dataset, it failed to learn an exponential decay time-dynamic pattern similar to $\Lambda$, overly relying on future time information. Both IPSU and "FPT w/ Masked $\mathbf{A}$" applied a lower triangular mask during training to ensure that future information was not utilized. These models focused more on the present and immediate past, neglecting older information. Notably, "FPT w/ Masked $\mathbf{A}$" learned a more distinct exponential decay pattern. For "FPT w/ $\mathbf{A}$", no constraints were imposed during training, resulting in a stronger focus on information from adjacent timesteps. The comparison of PSN with "FPT w/ $\mathbf{A}$" suggests that the reset mechanism favors learning reasonable

temporal patterns rather than maintaining reliance on distant temporal information.

## G. Ablation Study of $K$

*Table 9.* Effect of Different Iteration Counts $K$ on the DVS-Gesture Dataset.

|  | $K = 3$ | $K = 4$ | $K = 5$ |
|---|---|---|---|
| **DVS-Gesture** | 98.61 | 98.61 | 98.61 |

To evaluate whether increasing the number of iterations $K$ improves performance, we conducted experiments with different iteration counts under the same settings, including random seeds. The accuracy results on the DVS-Gesture dataset are shown in Table 9. As seen, increasing the iteration count does not affect accuracy. This is because, for $K \geq 2$, with $\alpha = 12$, the membrane potential $u_{(K)}$ converges by the time $K = 3$, and no significant changes occur with additional iterations. Therefore, the accuracy remains unchanged.

## H. Discussion and Limitation

**Motivation**: Our primary goal was to develop a parallel training approach that is independent of the model architecture, ensuring that the neuronal dynamics remain unchanged throughout the training process.

**Backward Process**: The forward process in FPT is identical to the original LIF model, ensuring that the key dynamics are retained. However, the backward process does not directly correspond to traditional BPTT.

**Reset Mechanism**: The reset mechanism is essential for certain tasks, especially those that require the network to focus on the current input rather than be influenced by previous states. For example, in tasks where the input remains constant over time, the reset mechanism might not be necessary. However, for tasks involving long-term dependencies or highly variable inputs, retaining the reset mechanism becomes crucial. It allows the network to discard irrelevant past information and focus on new, incoming data.

**Flexibility and Compatibility**: A key advantage of our algorithm is its flexibility. It imposes no restrictions on specific network architectures and can be applied broadly across various SNN models. Importantly, since FPT preserves the original neuron dynamics, models trained using FPT remain fully compatible with standard sequential SNN inference, allowing seamless deployment on existing neuromorphic hardware without modification. This compatibility, combined with its efficiency, makes FPT a practical and powerful approach for pretraining SNN models, bridging the gap between scalable training and real-world application.

**Limitations**: One limitation of FPT, compared to traditional SNN models, is its inability to handle scenarios where there is no time-dependent decay, as seen in Integrate-and-Fire models. However, more neuron models typically incorporate a decay factor or gating mechanism, which ensures that the influence of past inputs diminishes over time. This allows the model to focus on more recent data and prevents the accumulation of errors from earlier timesteps. Another limitation of FPT lies in the memory usage associated with parallel processing. Since FPT processes all timesteps simultaneously, it generally requires more memory than sequential training methods, as shown in Table 7. Future work could focus on optimizing memory efficiency. For instance, leveraging the binary nature of spike events could lead to more efficient memory compression techniques, enabling faster and more memory-efficient processing without sacrificing model performance.

