# OpenReview forum: "Efficient Parallel Training Methods for Spiking Neural Networks with Constant Time Complexity"
_ICML.cc/2025/Conference — ICML 2025 poster_

### Official Review · Reviewer_3oso · 2025-02-19

**Overall Recommendation:** 4

**Summary:**

This paper introduces a novel Fixed-point Parallel Training (FPT) method to accelerate Spiking Neural Networks (SNNs) training.
The method is theoretically analyzed for convergence, and the authors show that existing parallel spiking neurons are special cases of this approach.
Experimental results demonstrate that FPT simulates the dynamics of original LIF neurons effectively, significantly reducing computational time without sacrificing accuracy

**Claims And Evidence:**

Yes. The claim in this paper is clear.

**Essential References Not Discussed:**

No.

**Experimental Designs Or Analyses:**

Yes. Experiments in this paper is extensive and sound.

**Methods And Evaluation Criteria:**

Yes. The method is novel and theoretically complete.

**Other Comments Or Suggestions:**

1. The usage of "\cite" and "\citet" in latex should be careful. There are several misuses of "\cite" in section 2.3 and section 4.3.1
2. The authors should consider the running time comparison (or operation counts) of the training process and inference process between your method and traditional surrogate methods.

**Other Strengths And Weaknesses:**

Strengths:
1. correct and robust proofs;
2. well-written and clear motivations;
3. extensive and convincing experiments.

Weaknesses:
No obvious weaknesses.

**Questions For Authors:**

1. I wonder whether your method can be extended to pre-training SNNs parallelly.
There have been several works like SpikeBERT, SpikeGPT, and SpikeLM which pre-trained LIF-based spiking Transformers non-parallelly.
Please discuss the possibility of your method on pre-training SNNs.

**Relation To Broader Scientific Literature:**

This paper is an extension of previous work on parallel training SNNs.
I see this paper as a milestone because this method can be used to pre-train a SNN parallelly.

**Theoretical Claims:**

Yes. The proofs about fixed points are correct.

---

> ### Author Rebuttal · Authors · 2025-03-30
>
> We sincerely thank you for your high evaluation of our work, seeing this paper as a milestone due to its potential to enable parallel pre-training of SNNs. We also appreciate your valuable and detailed feedback. Below, we will answer your questions.
>
> **Q1: The usage of "\cite" and "\citet"**
>
> **A1:** We have carefully re-checked all citations throughout the manuscript. However, since ICML does not allow manuscript modifications during the rebuttal period, we will correct any remaining formatting issues (e.g., incorrect use of `\cite` vs `\citet`) in the camera-ready version if the paper is accepted.
>
> **Q2: Running time comparison of the training process and inference process**
>
> **A2:** Our proposed algorithm supports parallel training, including parallel forward and backward passes. We achieve speedups in both passes. To demonstrate this, we compare the training and inference complexity of BPTT and FPT at different timesteps ($T$). The experiments are conducted on the MNIST dataset using a 3-layer MLP (784×256×128×10), batch size = 256, learning rate = 0.001, and 80 training epochs.
>
> |Method|T|Training Time (s)|Inference Time (s)|Accuracy (%)|
> |-|-|-|-|-|
> |BPTT|8|0.0195|0.0042|$97.75\pm0.16$|
> ||64|0.0835|0.0257|$97.67\pm0.17$|
> ||512|1.701|0.2092|$97.84\pm0.12$|
> |FPT|8|0.0096|0.0021|$97.73\pm0.23$|
> ||64|0.0109|0.0021|$97.71\pm0.11$|
> ||512|0.0803|0.0021|$97.70\pm0.22$|
>
> Here, "time" refers to the average time required to train or infer a single batch on a single A100 GPU. As shown in the table, both the training and inference time of BPTT increase with the number of timesteps $T$. In contrast, the training time of FPT increases only slightly, and its inference time remains almost the same regardless of $T$. Notably, at $T = 512$, FPT is 21 times faster to train compared to BPTT.
>
> **Q3: Extended to pre-training SNNs parallelly**
>
> **A3:** A key advantage of our algorithm is its flexibility - it does not impose restrictions on specific network architectures and can be applied to a wide range of SNN models. By replacing the time-consuming LIF neuron sequential computation with our proposed FPT-based parallel iterations, both the forward and backward passes during training can be significantly accelerated.
>
> Importantly, since FPT preserves the original neuron dynamics, the pre-trained model can still be deployed and inferred using standard SNN sequential processing. This makes FPT not only suitable for accelerating training, but also highly compatible with existing SNN hardware deployments. Therefore, FPT provides a practical and general solution for efficient parallel pre-training of SNNs without sacrificing biological fidelity or inference compatibility.
>
> We again sincerely thank you for the thoughtful and encouraging comments. We hope that our response will clarify your concerns and further demonstrate the practicality and generality of FPT.

---

> > ### Comment · Reviewer_3oso · 2025-04-02
> >
> > I confirm my score. Good luck.

---

### Official Review · Reviewer_gXhg · 2025-03-11

**Overall Recommendation:** 3

**Summary:**

This work introduces Fixed-point Parallel Training (FPT), a novel method that reduces SNN training time complexity from O(T) to O(K) (where K is a small constant, typically K = 3) by enabling efficient parallel processing across all timesteps without modifying the network architecture. A theoretical convergence analysis proves the stability of FPT and demonstrates that existing parallel spiking neuron models are special cases of this framework. Importantly, FPT preserves LIF neuron dynamics, including membrane potential updates and reset mechanisms, ensuring both biological interpretability and computational efficiency. By decoupling sequential dependencies, FPT significantly accelerates training while maintaining or even improving accuracy, making it a scalable and efficient solution for large-scale, long-duration SNN tasks on modern hardware. This advancement enhances the feasibility of deploying SNNs in real-world applications, particularly in neuromorphic computing and time-sensitive spatiotemporal processing tasks.

**Claims And Evidence:**

The claims are supported by clear evidence.

**Essential References Not Discussed:**

No

**Experimental Designs Or Analyses:**

See "Methods And Evaluation Criteria"

**Methods And Evaluation Criteria:**

The "LIF Dynamics Simulation" part only include a single LIF neuron. However, the propagation of network dynamics might be quite different since the effect could accumulate. How does the network dynamics be affacted across long T.
How does the method perform on ImageNet-1K.

**Other Comments Or Suggestions:**

No

**Other Strengths And Weaknesses:**

See "Methods And Evaluation Criteria"

**Questions For Authors:**

See "Methods And Evaluation Criteria"

**Relation To Broader Scientific Literature:**

Yes

**Theoretical Claims:**

The proofs are correct.

---

> ### Author Rebuttal · Authors · 2025-03-30
>
> We thank you for your encouraging feedback and for recognizing that FPT significantly accelerates SNN training, making it a scalable and efficient solution for large-scale, long-duration tasks. We also appreciate your recognition of its potential in practical applications, especially neuromorphic computing and time-sensitive spatiotemporal processing. Below, we provide detailed responses to your questions and concerns.
>
> **Q1:  The network dynamics affacted across long T**
>
> **A1:** We replaced the LIF neurons in a pre-trained 3-layer LIF-based MLP (784×256×128×10) on the MNIST dataset with parallel LIF neurons based on FPT. The cosine similarity (%) between the original and FPT-replaced outputs for $T$=8, 64, and 512 is shown in the table below:
>
> ||T = 8|T = 64|T = 512|
> |-|-|-|-|
> |$\alpha=5$|99.89|99.55|99.53|
> |$\alpha=7$|99.90|99.49|99.47|
>
> As $T$ increases, the outputs of the FPT-based parallel LIF and the original LIF become less consistent due to error accumulation, resulting in a slight decrease in the similarity of the final network output. However, even for $T$=512, the similarity remains around 99.5%, indicating that FPT maintains a high degree of consistency in the network dynamics. Moreover, this minor difference can be addressed by light fine-tuning.
>
> **Q2: Experiments on ImageNet1K**
>
> **A2:** The proposed FPT mainly focuses on improving the speed for long $T$. Thus, in the submission, we reported results on dynamic datasets such as DVS-CIFAR10 ($T$=10) and DVS-Gesture ($T$=20), static datasets including ImageNet-100 ($T$=4), and graph datasets like Amazon Photos ($T$=8, 32, 128), as well as sequential datasets such as Sequential CIFAR10 ($T$=32) and Sequential CIFAR100 ($T$=32). In contrast, static datasets like ImageNet usually use smaller timesteps (e.g., $T$=5 or 6).
>
> In addition, due to the high computational cost of training SNNs on ImageNet-1K, many previous studies (e.g., SSNN and LocalZO) have adopted alternative benchmarks such as Tiny-ImageNet or ImageNet-100. For this reason, we also chose ImageNet-100 as the representative benchmark in the main results. We are currently actively conducting experiments on ImageNet-1K, using the same experimental settings as MS-ResNet-18 [1]. Due to time constraints, we report the preliminary results we have obtained so far below.
>
> ||BPTT (82 epoch, our running, lastest)|FPT (82 epoch, our running)|FPT (97 epoch, lastest)|
> |-|-|-|-|
> |Accuarcy (%)|52.27|53.57|57.55|
> |Training loss|2.21|2.18|1.98|
> |Per-batch time (s)|8.48|5.76|5.76|
>
> The training loss refers to the cross-entropy loss of the last batch in the given epoch. As seen, with the same hyperparameters and number of epochs, FPT achieves higher accuracy and lower loss. Additionally, even with $T$=6, FPT achieves approximately 1.5× faster training speed compared to BPTT on the same 4 3090 GPUs. It is important to note that, due to time and computational limitations, we have not yet fine-tuned the hyperparameters for FPT. Further adjustments may lead to even better performance.
>
> [1] Advancing Spiking Neural Networks Toward Deep Residual Learning. TNNLS 2024
>
> We sincerely hope that our clarifications have addressed your concerns and will help improve your opinion of our work.

---

### Official Review · Reviewer_rhny · 2025-03-13

**Overall Recommendation:** 2

**Summary:**

The paper proposes a new training method for SNNs called Fixed-point Parallel Training (FPT), which aims to improve efficiency by reducing time complexity from O(T) to O(K), where K is a small constant. The method leverages a fixed-point iteration framework to enable parallel computation across timesteps rather than processing them sequentially. The authors argue that this approach preserves key neural dynamics, including the reset mechanism of Leaky LIF neurons, while accelerating training significantly. Theoretical analysis is provided to prove the convergence of FPT, and the authors demonstrate that some existing parallel spiking neuron models can be viewed as special cases of this method. Experiments show that FPT achieves competitive accuracy while significantly reducing computational costs compared to traditional BPTT and other existing parallel spiking neuron training methods.

## update after rebuttal

Thanks to the authors for the detailed and thoughtful rebuttal. I appreciate the additional clarifications on memory usage, the effect of surrogate gradients on convergence, and the broader applicability of FPT. It’s clear a lot of effort went into addressing the points I raised.

That said, my overall assessment remains the same. While the new details help, the broader concerns — like the limited experimental scope, lack of deeper analysis on the practical trade-offs, and the fairly narrow range of tested applications — are still there. I think the paper presents a promising idea and the results are solid within the datasets tested, but it doesn't quite reach the level of novelty and thorough validation I’d expect for acceptance. So I’m keeping my original score of Weak Reject.

**Claims And Evidence:**

The main claim is that FPT allows parallel training of SNNs without modifying the network architecture while maintaining biological interpretability. The authors provide both theoretical and empirical support for this claim. The convergence proof of the fixed-point iteration is a positive contribution, though it relies on certain assumptions about the Lipschitz continuity of the surrogate function. The empirical results show that FPT achieves a significant speedup over BPTT while maintaining comparable accuracy across multiple datasets. However, the experiments lack detailed comparisons with alternative approaches such as event-based SNN training techniques or alternative parallelization strategies like recurrent SNNs with gating mechanisms. The claim that FPT generalizes well to different datasets is only partially supported since the paper primarily focuses on neuromorphic datasets like DVS-CIFAR10 and DVS-Gesture but does not test on more complex real-world applications, such as continuous control or large-scale spiking datasets.

One aspect that is not thoroughly examined is the potential trade-off between computational speed and memory usage. Since FPT processes all timesteps in parallel, it likely requires more memory than sequential training methods. The authors acknowledge this in the discussion but do not provide a quantitative breakdown of memory overhead. The claim that FPT maintains all critical neural dynamics is also somewhat oversimplified because the backward process does not exactly correspond to traditional BPTT. The experimental results support the main claims to some extent, but the paper would benefit from deeper analysis of efficiency trade-offs and broader comparisons to alternative SNN training methods.

**Essential References Not Discussed:**

The paper discusses prior work on parallel SNN training and timestep reduction but does not cite some relevant research on alternative acceleration techniques, such as event-driven training methods or hybrid ANN-SNN approaches. There is also limited discussion of how FPT compares to neuromorphic hardware implementations, which are a key motivation for efficient SNN training. Some recent work on efficient surrogate gradient methods and biologically inspired training rules could also be relevant for positioning FPT within the broader field.

**Experimental Designs Or Analyses:**

The experimental setup is well-structured, with comparisons across multiple datasets and an ablation study to analyze key design choices. The main advantage demonstrated is the speedup of training, particularly for long-duration simulations where traditional BPTT is inefficient. However, there are some weaknesses in the analysis. The paper does not include a detailed breakdown of computational cost beyond training time, such as memory usage and GPU utilization. While FPT reduces the number of sequential operations, it likely increases parallel memory requirements, and this trade-off is not analyzed in depth.

Another issue is the limited scope of dataset selection. Most of the experiments focus on neuromorphic benchmarks, which are useful for validating the method but do not fully demonstrate its applicability to broader machine learning tasks. It would be beneficial to see results on tasks like speech recognition, reinforcement learning, or other domains where SNNs are increasingly being applied. Finally, the statistical significance of the results is not clearly reported. While standard deviations are included, there is no discussion of whether the differences in accuracy are statistically significant across multiple runs.

**Methods And Evaluation Criteria:**

The authors evaluate FPT on standard neuromorphic datasets such as DVS-CIFAR10, DVS-Gesture, and ImageNet-100. These datasets are commonly used for benchmarking SNNs, so the choice is reasonable. The experiments compare FPT to existing training methods such as timestep shrinkage, online training, and stabilized spiking flow, providing a fair assessment of performance gains. However, the evaluation criteria focus almost entirely on accuracy and training speed, without considering other important factors like memory usage, energy efficiency, or sensitivity to hyperparameters. Since one of the key motivations of SNNs is energy efficiency, a discussion of power consumption during training would be valuable.

The authors also conduct an ablation study to examine the impact of the number of iterations (K) and the role of the reset mechanism. This is a strong aspect of the evaluation, as it provides insight into the effectiveness of the method. However, a more detailed breakdown of how K affects convergence speed and accuracy would make the evaluation even stronger. The experimental design is generally sound but could be improved by including a broader set of tasks, particularly those requiring long-term dependencies, to further validate the generalization ability of FPT.

**Other Comments Or Suggestions:**

The authors should include a discussion on memory efficiency and computational cost beyond training time. They should also test FPT on a broader range of tasks to demonstrate its applicability. Providing open-source code would improve reproducibility.

**Other Strengths And Weaknesses:**

The main strength of the paper is its focus on improving the efficiency of SNN training without modifying the underlying network architecture. The use of fixed-point iteration for parallel training is a novel and well-motivated approach. The experimental results demonstrate clear speed improvements over BPTT while maintaining accuracy. However, the paper has several weaknesses. The evaluation lacks a detailed analysis of computational trade-offs, particularly regarding memory usage and energy efficiency. The theoretical analysis, while useful, does not fully address the impact of surrogate gradients on convergence. The experiments are mostly limited to neuromorphic datasets, which may not fully demonstrate the generalization ability of FPT.

**Questions For Authors:**

1. How does FPT compare in memory consumption to standard BPTT?
2. How does the surrogate gradient approximation affect convergence guarantees?
3. Can FPT be applied to reinforcement learning tasks or other non-spiking domains?
4. What are the potential failure cases where FPT might not converge effectively?

**Relation To Broader Scientific Literature:**

The paper is well-grounded in the existing literature on SNN training methods and parallel computing in neural networks. It references prior work on timestep reduction, online training, and surrogate gradient methods, positioning FPT as an improvement over these approaches. However, the discussion is somewhat limited to SNN-specific methods and does not draw connections to broader machine learning literature. There are similarities between FPT and parallelization techniques used in recurrent neural networks (RNNs) and deep equilibrium models, but these connections are not explored in depth. It would be useful to compare FPT to other parallel training techniques used in non-spiking networks to highlight its broader relevance.

**Theoretical Claims:**

The paper provides a theoretical proof that FPT converges to a fixed point under certain conditions. This is an important contribution, as many SNN training methods lack formal convergence guarantees. The proof is based on contraction mapping and Lipschitz continuity arguments, which are reasonable assumptions for neural networks with smooth surrogate gradients. However, the proof does not establish how the convergence rate of FPT compares to BPTT or whether it guarantees optimal weight updates in all cases. The surrogate gradient approach used in the backward pass introduces an additional layer of approximation, and the impact of this approximation on convergence is not fully analyzed. The theoretical claims are generally well-supported but would benefit from additional discussion on potential failure modes, such as scenarios where the fixed-point iteration might converge too slowly or to suboptimal solutions.

---

> ### Author Rebuttal · Authors · 2025-03-31
>
> We sincerely thank you for recognizing that FPT enhances the efficiency of SNN training without modifying the underlying network architecture and for acknowledging FPT as a novel and well-motivated approach. Below, we provide detailed responses to your questions.
>
> **Q1: How does FPT compare in memory consumption to standard BPTT?**
>
> **A1:** To compare the memory consumption of BPTT and FPT, we conducted experiments on the MNIST dataset using a 3-layer MLP (784×256×128×10), a batch size of 256, a learning rate of 0.001, and 80 epochs. The table below shows the comparison at different $T$:
>
> |Method|T|Training Time (s)|Inference Time (s)|Memory (MB)|Accuracy (%)|
> |-|-|-|-|-|-|
> |BPTT|8|0.0195|0.0042|600|$97.75\pm0.16$|
> ||64|0.0835|0.0257|956|$97.67\pm0.17$|
> ||512|1.701|0.2092|3238|$97.84\pm0.12$|
> |FPT|8|0.0096|0.0021|622|$97.73\pm0.23$|
> ||64|0.0109|0.0021|1162|$97.71\pm0.11$|
> ||512|0.0803|0.0021|4264|$97.70\pm0.22$|
>
> Here, "Time" refers to the average running time for one batch during training or inference on a single A100 GPU. At $T=512$, FPT is 21x faster than BPTT during training.
>
> FPT only slightly increases the memory usage for the LIF during training without affecting other components of the network.
> Thus, the overall memory consumption is not significantly higher than BPTT. For instance, in larger networks such as MS-ResNet-18 trained on ImageNet1K, BPTT consumes 22.54 GB on 4 3090 GPUs, while FPT consumes 23.52 GB, resulting in only a 4% increase in memory usage.
>
> **Q2: How does the surrogate gradient approximation affect convergence guarantees?**
>
> **A2:** The surrogate gradient approximation does not affect the convergence guarantees because these guarantees are based on the forward pass of the network, while the surrogate gradient approximation is applied on the backward pass. In addition, the surrogate gradient approximation is necessary for backpropagation in SNNs due to the non-differentiability of the spike activation function. The effectiveness of surrogate gradient approximation has been well verified through experiments on various datasets.
>
> **Q3: Can FPT be applied to reinforcement learning tasks or other non-spiking domains?**
>
> **A3:** FPT can indeed be applied to reinforcement learning tasks as long as the task uses LIF-based SNNs for temporal processing. Our approach is particularly suitable for addressing the high latency problem of sequence processing using LIF neurons. However, for other non-spiking domains that do not rely on SNNs, this is beyond the scope of our current work.
>
> **Q4: What are the potential failure cases where FPT might not converge effectively?**
>
> **A4:** In the submission，we have discussed this issue in Appendix F "Discussion and Limitation". FPT works well for LIF-based SNNs with temporal decay, but may not be effective for simple Integrate-and-Fire models that lack such mechanisms. However, mainstream neuron models typically incorporate a decay factor or gating mechanism, which ensures that the influence of past inputs diminishes over time.
>
> **Other Questions**
>
> - *The claim that FPT maintains all critical neural dynamics is also somewhat oversimplified because the backward process does not exactly correspond to traditional BPTT.*
>
> Neural dynamics, including leakage, integration, firing, and reset, are part of the neuron's forward process. Neither the backward process of BPTT nor FPT belongs to the neural dynamics process; they are solely used to optimize network weights.
>
> - *Since one of the key motivations of SNNs is energy efficiency, a discussion of power consumption during training would be valuable.*
>
> The energy efficiency of SNNs typically refers to inference on neuromorphic hardware. Both BPTT and FPT rely on floating-point operations on GPUs during training, which does not reflect deployment energy costs. FPT does not affect sequential inference behavior of SNNs. As shown in Table 2 of the submission and in additional experiments presented in **Review hARo (A1)**, its firing rate is comparable to or even lower than BPTT, ensuring that SNNs trained with FPT remain energy efficient during deployment.
>
> - *The paper primarily focuses on neuromorphic datasets like DVS-CIFAR10 and DVS-Gesture*
>
> In the submission, we reported results on dynamic datasets such as DVS-CIFAR10 and DVS-Gesture, static datasets including ImageNet-100, and graph datasets like Amazon Photos, as well as sequential datasets such as Sequential CIFAR10 and Sequential CIFAR100.
>
> - *Additional numerical experiments demonstrating different convergence behaviors would strengthen the argument.*
>
> In the submission, Section 6.1 "LIF Dynamics Simulation" already discusses the convergence behavior.
>
> - *Providing open-source code would improve reproducibility.*
>
> The code will be publicly available on GitHub after the publication of this work.
>
> We sincerely hope that our clarifications above have addressed your concerns and that our responses contribute positively to your understanding of our work.

---

### Official Review · Reviewer_hARo · 2025-03-14

**Overall Recommendation:** 4

**Summary:**

This paper proposes Fixed-point Parallel Training for efficient training of SNN, which does not change the network architectures. This training mode does not affect the dynamics of LIF neurons and achieves better performance on data with time series information such as DVS.

**Claims And Evidence:**

The three contributions of this paper are:

- It "proposes a novel Fixed-point Parallel Training (FPT) method that reduces the training time complexity of SNNs";
- it "proves the convergence of FPT and demonstrates that existing parallel spiking neuron models can be derived as special cases of our framework";
- it "retains the dynamic properties of original LIF neurons, achieves better performance, and significantly reduces computational time".

I agree with the first two contributions, but I am skeptical about the last one.

- This paper does not show in detail the computation, time, and space complexity of using FPT to train SNN. Please refer to T-RevSNN [1] for detailed experimental data on various complexities during training.
- FPT is mainly experimented on time series data, lacking experiments on static data. Please add experiments on ImageNet1k to prove that FPT is still reliable on static data tasks in the rebuttal.

[1] High-Performance Temporal Reversible Spiking Neural Networks with O(L) Training Memory and O(1) Inference Cost. ICML 2024.

**Essential References Not Discussed:**

N/A

**Experimental Designs Or Analyses:**

Please see `Methods And Evaluation Criteria`.

**Methods And Evaluation Criteria:**

This paper lacks experiments on large static datasets, such as ImageNet1k. Also, the DVS data selected in this article is also relatively small. If possible, please consider supplementing the experiments on HAR-DVS [1] in the rebuttal.

[1] HARDVS: Revisiting Human Activity Recognition with Dynamic Vision Sensors. AAAI 2024.

**Other Comments Or Suggestions:**

There is no description of the `Time` in Table 2 and Figure 3, which may confuse me.

**Other Strengths And Weaknesses:**

N/A

**Questions For Authors:**

Please see `Claims And Evidence`, `Methods And Evaluation Criteria`, and `Supplementary Material`.

**Relation To Broader Scientific Literature:**

This paper focuses on addressing parallel training methods for SNNs and does not seem to make an obvious contribution to the broader scientific literature.

**Theoretical Claims:**

The proof in Appendix A is correct in my view.

---

> ### Author Rebuttal · Authors · 2025-03-30
>
> We sincerely thank the reviewers for recognizing the novelty and contributions of our proposed FPT framework, including its ability to reduce the training complexity of SNNs and the fact that existing parallel spiking neuron models can be viewed as special cases of FPT. We also appreciate the constructive and insightful feedback. Below, we answer the reviewers' questions in detail.
>
> **Q1: Various complexities of FPT during training refer to T-RevSNN**
>
> **A1**: Thank you for your valuable suggestion to further demonstrate the various complexities of FPT. The table below compares the theoretical training and inference complexity of different algorithms:
>
> |Methods|Training||Inference|Applicable scope|
> |-|-|-|-|-|
> ||Memory|Time|Energy||
> |OTTT|$O(L)$| $O(LT)$|$O(T)$|Limited|
> |SLTT-k|$O(L)$|$O(Lk)$|$O(T)$|Limited|
> |T-RevSNN turn-off|$O(L)$|$O(L)$|$O(1)$|Limited|
> |T-RevSNN turn-on|$O(L)$|$O(T)$|$O(1)$|Limited|
> |BPTT|$O(LT)$|$O(LT)$|$O(T)$|Unlimited|
> |FPT|$O(LT)+\lambda O(LKT)$|$O(LK)$|$O(T)$|Unlimited|
>
> Here, $L$ is the number of network layers, $T$ is the timestep, and $K$ is the number of iterations in FPT, which is typically 3 and does not increase with $T$. Thus, for long $T$, $K$ can be approximated to be negligible. The space complexity $O(LKT)$ and time complexity $O(LK)$ can be approximated as $O(LT)$ and $O(L)$, respectively. The coefficient $\lambda$ represents the proportion of memory attributed to LIF components, which is the only part FPT increases—other parts of the network remain unaffected.
>
> It is worth noting that methods such as OTTT, SLTT, and T-RevSNN truncate gradients or discard most of the temporal connections, which can limit their applicability to tasks requiring fine-grained temporal dynamics. In contrast, FPT accelerates SNN training without modifying the network and retaining the original neuron dynamics, so it is applicable to a wider range of SNN models.
>
> Due to time constraints and the limitations of OTTT, SLTT, and T-RevSNN in capturing temporal features, our experiments mainly focus on comparing the complexity of BPTT and FPT at different timesteps $T$. The experiments are conducted on the MNIST dataset using a 3-layer MLP (784×256×128×10), a batch size of 256, a learning rate of 0.001, and 80 epochs.
>
> |Method|T|Training Time (s)|Inference Time (s)|Firing rate (%, layer 1)|Firing rate (%, layer 2)|Memory (MB)|Accuracy (%)|
> |-|-|-|-|-|-|-|-|
> |BPTT|8|0.0195|0.0042|44.47|47.03|600|$97.75\pm0.16$|
> ||64|0.0835|0.0257|45.1|46.43|956|$97.67\pm0.17$|
> ||512|1.701|0.2092|46.55|47.76|3238|$97.84\pm0.12$|
> |FPT|8|0.0096|0.0021|40.83|44.29|622|$97.73\pm0.23$|
> ||64|0.0109|0.0021|38.24|40.36|1162|$97.71\pm0.11$|
> ||512|0.0803|0.0021|38.14|39.94|4264|$97.70\pm0.22$|
>
> Here, "Time" refers to the average running time for one batch during training or inference on a single A100 GPU. As shown in the table, both training and inference time for BPTT increase with $T$. In contrast, for FPT, the training time increases slightly, while the inference time remains almost constant. At $T=512$, FPT is 21x faster than BPTT during training. FPT only slightly increases the memory usage for the LIF during training without affecting other components of the network. In larger networks, such as MS-ResNet-18 trained on ImageNet1K, BPTT occupies 22.54GB on 4 3090 GPUs, while FPT occupies 23.52GB, resulting in only about a 4% increase in memory usage.
>
> **Q2: Experiments on ImageNet1K and If possible, HAR-DVS**
>
> **A2:** For a detailed discussion of the ImageNet1K experiments, we respectfully refer you to our response to **Reviewer gXhg (A2)**, due to space constraints here.
>
> We thank the reviewer for suggesting the HAR-DVS dataset, a promising new benchmark for DVS-based action recognition. We will cite it in the introduction: `For instance, neuromorphic benchmark datasets such as HAR-DVS, DVS-CIFAR10 and DVS-Gesture typically need 10 or more timesteps to reach satisfactory accuracy.` However, due to time and computational constraints, and because this dataset was recently released, the main algorithms we compare have not yet reported results on it, making a direct comparison difficult in the short term.
>
> **Q3: TET Loss**
>
> **A3:** The baseline accuracy and metrics in our submission are from the best results in the respective papers. As TET loss is a mainstream loss function for SNNs, most of these baselines, such as T-RevSNN and LocalZO, utilize TET loss during training.
>
> **Q4: "Time" in Table 2 and Figure 3**
>
> **A4:** We apologize for any confusion. The "time" here refers to the average time required to train one batch on a single 3090 GPU. All baseline models were implemented with the same network architecture and hyperparameter configuration as ours, differing only in training method. FPT requires significantly lower training time than these baselines.
>
> We sincerely hope that our clarifications have addressed your concerns and helped strengthen your confidence in our work. Thank you again for your thoughtful review.

---

> > ### Comment · Reviewer_hARo · 2025-04-03
> >
> > The rebuttal addresses my issues and I will raise the score to 4.

---

### Decision · Program_Chairs · 2025-05-01

**Decision:**

Accept (poster)

**Comment:**

This paper proposes a novel fixed-point parallel training method for spiking neural networks, aiming to accelerate training by reducing temporal complexity without modifying the network architecture. The method simulates the dynamics of LIF neurons and achieves improved performance, particularly for long-duration simulations.

Reviewer hARo acknowledged the theoretical contributions. Reviewers rhny and gXhg found the experiments with speed improvements while maintaining accuracy. Reviewer 3oso further acknowledged the theoretical analysis and the extensiveness of experiments for scalable parallel SNN training.

Reviewer hARo questioned the generality of the method on large-scale datasets and insufficient discussion of training complexity. Reviewer rhny also raised concerns about comparisons with other models. Reviewer gXhg suggested a more detailed explanation of the method. Besides, Reviewer 3oso suggested minor improvements in citation formatting. The authors addressed their concerns during the rebuttal phase. Following the discussion, the paper received final scores of (4, 4, 3, 2). Overall, I think this work is technically sound and useful. Therefore, I recommend this paper be accepted.